# Benthic foraminifera as bio-indicators of natural and anthropogenic conditions in Roscoff Aber Bay (Brittany, France)

**Edwin Daché[1]\*, Pierre-Antoine Dessandier[1], Ranju Radhakrishnan[1], Valentin Foulon[2], Loïc Michel[3], Colomban de Vargas[4], Jozée Sarrazin[1], Daniela Zeppilli[1]**

**1** Ifremer, BEEP, Univ Brest, Plouzané, France, **2** ENIB - École Nationale d'Ingénieurs de Brest, Plouzané, France, **3** Laboratory of Oceanology, Freshwater, and Oceanic Sciences Unit of reSearch (FOCUS), University of Liège, Liège, Belgium, **4** CNRS, Station Biologique de Roscoff, AD2M, UMR 7144, ECOMAP, Sorbonne Université, Roscoff, France

\* Edwin.Dache@ifremer.fr

**Data Availability Statement:** All relevant data are within the manuscript and its Supporting Information files.

## Abstract

Living benthic foraminifera, known as environmental bio-indicators of both natural and anthropogenic conditions in marine environments, were investigated in the coastal environment of Roscoff Aber Bay (Brittany, France). Eight sampling sites subject to natural variations (freshwater inputs, tides) and/or anthropogenic impacts (pollution, eutrophication) were studied over four seasons in 2021–2022 (November, February, May, August). We sought to understand the spatial distribution of foraminiferal populations within and between sampling sites over the different seasons and to identify sensitive species and those tolerant to anthropogenic impacts. To this end, sedimentary and biogeochemical characteristics of the sediments were examined by measuring grain size, temperature, oxygen, salinity, pH, environmental pigment concentration (chl $a$ and phaeopigments), total organic carbon (TOC), isotopic ratios of carbon ($\delta^{13}C$), nitrogen ($\delta^{15}N$) and sulfide ($\delta^{34}S$), and chl $a$ fluorescence. Considering these parameters as potential driving factors, four environments were distinguished among the sampling sites: open water, terrestrial, oligotrophic and eutrophic. These showed an increasing gradient of organic supply as well as very different microbial activities, highlighted by carbon and sulfide isotopic ratios. Foraminiferal population study revealed the dominant species characterising these main environments. The lowest abundance but highest diversity of foraminifera was found in the harbour site, associated with the dominance of *Haynesina germanica*, suggesting this species is tolerant to eutrophic environments and anthropogenic impacts. Open water was dominated by *Ammonia beccarii* and *Elphidium crispum*, while *Quinqueloculina seminula* was the most abundant species in the site with the greatest terrestrial influence. Interestingly, the observed organic enrichment of the harbour due to anthropogenic activities (fisheries, waste deposits, etc.) does not seem to significantly affect foraminiferal diversity. Overall, the benthic foraminiferal species in Roscoff Aber Bay appear to be an excellent proxy for marine environmental conditions under various natural and anthropogenic influences.

**Funding:** This work was supported by the BLUE REVOLUTION project (Biodiversity underestimation in our bLUe planEt: artificial intelligence REVOLUTION in benthic taxonomy) funded by the Interdisciplinary Graduate School for the Blue Planet (ISBlue; ANR-17-EURE-0015) and Ifremer (Institut français de recherche pour l'exploitation de la mer). DZ was supported by the project "Massive mEIOfauna DiscoverY of new Species of our oceans and SEAs (MEIODYSSEA) funded by the Sasakawa Peace Foundation. RR was supported by the ISA-IFREMER Collaboration in support of the capacity development of national from developing States, by the Ifremer Marine Mineral Resources project (REMIMA project) and by the French National Research Agency under France 2030 (reference ANR-22-MAFM-0001).

**Competing interests:** The authors declare that they have no conflicts of interests.

## Introduction

Foraminifera are a highly abundant phylum that dominate the meiobenthos as the most diverse group of shelled organisms [1]. A short life cycle and broad spatial distribution make them sensitive to environmental conditions as they respond rapidly to both natural and anthropogenic changes [2, 3], playing a major role in organic matter cycling [4, 5]. These characteristics make them relevant bioindicators for environmental quality or proxies to monitor environmental changes [2]. Previous studies have shown foraminifera to be good model organisms to study anthropogenic impacts in coastal areas, such as pollution in harbours [6], organic matter accumulation [7], thermal pollution [8] or oil accidents [9]. The Foram-AMBI index [10] has been used to identify groups ranging from non-sensitive to tolerant and opportunistic species. The standardized protocol established in 2011 by the FOBIMO group [11] allowed the responses of foraminiferal species to be identified in different study areas, including the Mediterranean Sea [12], Arctic fjords [10], intertidal zones [13] and estuaries [14].

Monitoring environmental changes in marine systems can be hindered by a multitude of influencing factors. Intertidal environments are impacted by stresses of both natural (e.g., tidal regime, meteorological and hydrodynamic variations, salinity gradients, temperature changes, sediment types and chemistry) and anthropogenic (e.g. eutrophication, contamination by heavy metals and chemicals, oil pollution and thermal impacts of power plants) origin [2, 15], making the evaluation of human impacts difficult to disentangle [16]. Organic carbon concentration is widely used as a proxy for eutrophication, often associated with the accumulation of pollutants and contaminants in coastal areas [10]. Intertidal areas are also characterised by strong seasonal variability that is often neglected due to the high sampling effort required for faunal and environmental analyses in each season. This underlines the necessity for additional studies in environmental monitoring, utilizing indices that incorporate comprehensive datasets and account for seasonal variations in these environments.

The natural variability of environmental conditions in Roscoff Aber Bay, located on the French northwest coast, has already been described in numerous studies [17–19]. The bay is a large flat-bottomed depression above the mid-tide line [20], characterised by habitats with very different sediment grain sizes, strong variations in sea height due to tidal influence and freshwater inputs influencing salinity [21]. At the exit of the bay, a channel is used intensively by boats going to nearby Batz Island. Between the two, the harbour is impacted by inputs of organic matter linked to fishing and hydrocarbon spills. Two monitoring stations (one inshore, the other offshore) collect data on numerous environmental and biological parameters [22], thus providing descriptions of environmental conditions in this dynamic environment.

Historically, the Roscoff Biological Station was established in this area because of the high species diversity. Indeed, coastal plankton [23, 24], algae [25–27] and macro-organisms [28–30] have all now been intensively studied in the Roscoff region where they show high biodiversity and habitat variability. This area is also characterised by human influences such as tourism and fishing activities, for which the impact on the meiobenthos remains poorly described. Monitoring of benthic foraminifera as bio-indicators has notably never been carried out in this area. This raises questions on how the foraminiferal community responds to anthropogenic impacts and how we can discriminate this response from natural environmental variation in the area. These questions are the focus of the present study. Additionally, we aim to investigate the use of foraminiferal species as bio-indicators for seasonal environmental changes.

In this context, we analysed the spatial and seasonal distribution of living benthic foraminifera from contrasting habitats impacted by natural and anthropogenic environmental changes,

with the aim of identifying bio-indicator species. To assess the seasonal and spatial variabilities, sampling was conducted in November, February, May and August at eight sites in the inner bay and the outer channel as well as in the harbour. We focused on the distribution and abundance of living foraminiferal species in order to determine their ecology and identify those species sensitive or tolerant to environmental changes, ultimately comparing our results with existing ecological indices.

## Materials and methods

### Study area

The study area is located on the coast off Roscoff (Brittany, France), in and around a small bay of 2 km$^2$. The bay consists of a small cove 2 km long and 1 km wide, partly silted up with different types of intertidal sediment [31]. A freshwater stream enters the cove and communicates between the polder and the sandy-muddy north-eastern region. The cove is very shallow, located completely above the mean tide level, with strong currents and a tidal range of about 4 metres, allowing complete mixing and high turbidity [20]. A series of eight sampling sites were selected, differently exposed to anthropogenic impacts. Stations 1–5 are located in the subtidal zone of the bay, station 5 being nearest to a freshwater stream (Fig 1). Station 6 is located in the old fishing port of Roscoff (Fig 1). Stations 7 and 8 are in subtidal zones located outside the bay and subject to strong channel currents between Batz Island and Roscoff (Fig 1). Environmental and faunal sampling was done in November 2021, February 2022, May 2022 and August 2022.

### Sample collection

Quadrats of 30 cm × 30 cm were deployed at each of the eight stations and a Plexiglas corer of 3 cm diameter and 5 cm height, representing a volume of 35 cm$^3$, was used for sampling [32]. Three cores were taken per station within the quadrat to analyse granulometric, pigmentary and isotopic parameters, giving a total of 24 cores per season. These cores were frozen at -20˚C. Another three sediment cores per station were also taken for subsequent analysis of the associated meiofauna. In this study, only a single core was used to determine living benthic foraminifera (by phloxine B staining) [33]. The sediment cores were first immersed in 6% $MgCl_2$ for 10 minutes before fixation with 4% borax-buffered formalin to allow extraction of the meiofauna from the sediments by centrifugation with LUDOX® colloidal silica [32]. To help with data analysis, a SOMLIT (Service d'Observation en Milieu Littoral) sampling station located near our study area provided regular data on environmental parameters [34]. We were able to plot salinity, oxygen, pH and chlorophyll *a* (Chl *a*) data for the period of November 2021 to August 2022 (S1 Fig).

### Sedimentary and geochemical analyses

**Grain size analysis.**  Particle size analysis was carried out with a Malvern™ Mastersizer 3000 laser diffractometer, which has a measurement range of 0.01–2200 μm. The measurements were performed on the entire sediment core. In this study, only the statistical mode Q 50 is represented, corresponding to the average particle size of each sampling station [35].

**Oxygen, pH, temperature and salinity measurements.**  At all stations, salinity parameters were measured with an LF 340 handheld conductivity meter with a standard TetraCon 325 conductivity cell (Measuring Range 1 μS/cm—2 S/cm) and pH parameters were measured with a WTW pH 3310 sensor (accuracy ± 0.005). Temperature and oxygen measurements were made with a Oxygen Optode 3830 (temperature accuracy ±0.05˚C and $O_2$-concentration

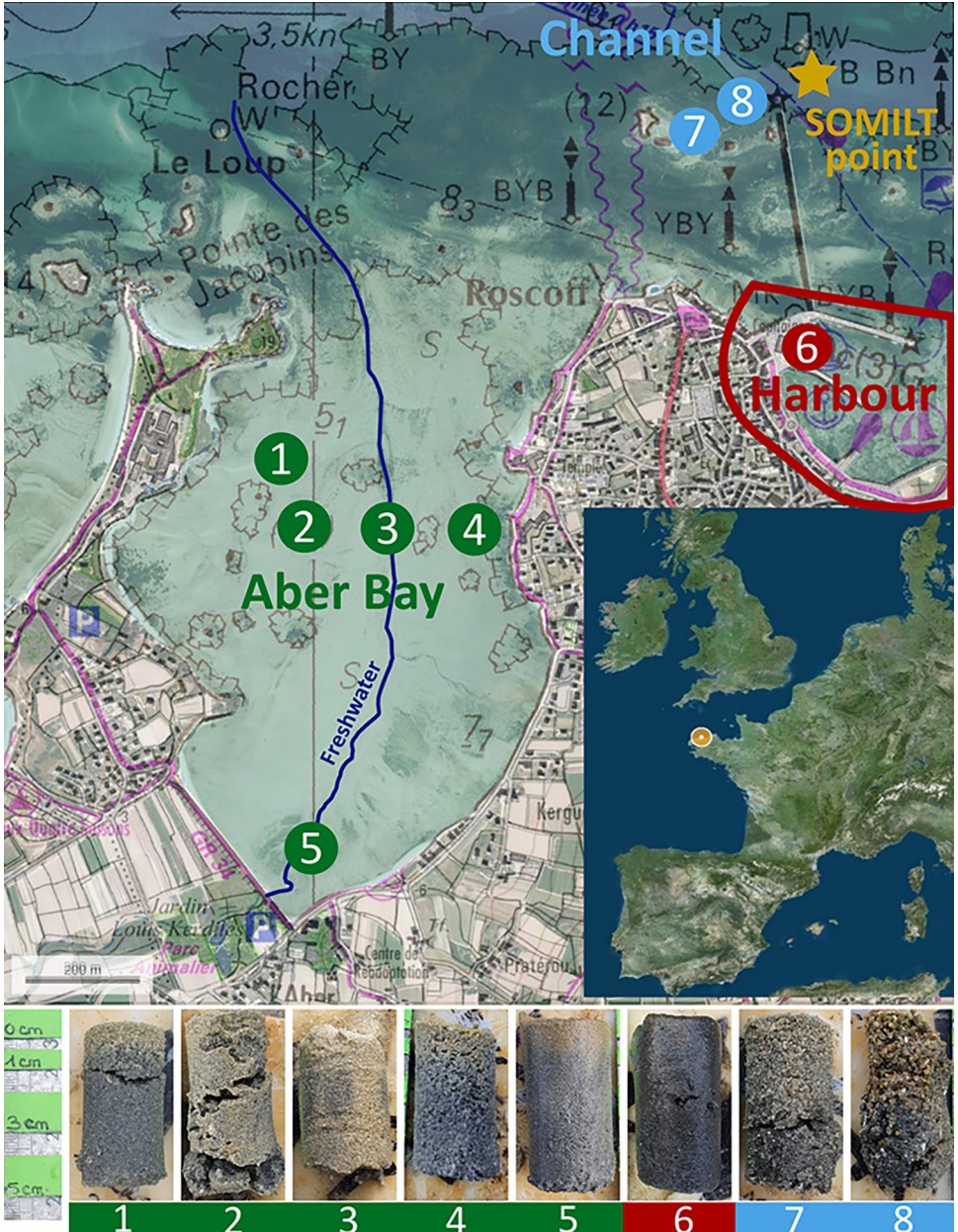

**Fig 1. Satellite map showing the positions of the different Roscoff Aber Bay sampling stations.** Stations 1–5 inside the bay, station 6 in the harbour and stations 7–8 outside the bay.

accuracy < 8 µM or 5%, whichever is greater). Measurements were taken directly during core sampling in a hole in the sediment containing interstitial seawater. Once the measurement had stabilised, the maximum value was taken. This procedure was repeated throughout the seasons.

**Pigment analysis.** Photosynthetic pigments were analysed on the entire sediment core. The pigments were extracted with 99.9% methanol solvent. The supernatant was centrifuged several times to avoid sediment entering the high-performance liquid chromatography (HPLC) system. Then, 170 µL of the extract and 30 µL water were mixed and a 100 µL aliquot

of this mixture immediately injected into the HPLC system. Filtration, extraction procedures and HPLC pigment analyses were performed following Zapata *et al.* [36]. Total organic carbon (TOC) and chlorophyll *a* (Chl *a*) indicate the total amount of organic matter (OM) and phyto-detritus, while the quality of OM was assessed using the ratio of Chl *a* to phaeopigments (Chl *a*/Phaeo), which indicates the freshness of the phytodetrital material [35].

**Elemental and isotopic analyses.**   Sediment samples were freeze-dried at -50˚C and ground to a fine powder using a grinder (model mixer MM 400). Sediment aliquots were ground to a homogeneous powder using a mortar and pestle, acidified to remove carbonates by direct addition of excess 1M HCl in small increments, and then rinsed with distilled water. Sediment samples were analysed twice: once using acidified material (for total organic carbon content and carbon stable isotope ratios) and once using native material (for total carbon content, total nitrogen and sulfur content, and nitrogen and sulfur stable isotope ratios).

Elemental content was measured using a vario MICRO cube C-N-S elemental analyser (Elementar Analysensysteme GMBH, Hanau, Germany) as relative percentage of analysed mass (mass%). Empty tin cups were used as analytical blanks. Sulfanilic acid (Sigma-Aldrich; %C = 41.6%, %N = 8.1%, %S = 18.5%) was used as the elemental standard.

Stable isotope ratio measurements were performed via continuous flow-elemental analysis-isotope ratio mass spectrometry (CF-EA-IRMS) at the University of Liège (Belgium), using the abovementioned vario MICRO cube C-N-S elemental analyser coupled to an IsoPrime100 isotope ratio mass spectrometer (Isoprime, Cheadle, United Kingdom). Isotopic ratios were expressed using the conventional δ notation [37], in ‰ and relative to the international references Vienna Pee Dee Belemnite (for carbon), atmospheric air (for nitrogen) and Vienna Canyon Diablo Troilite (for sulfur). IAEA (International Atomic Energy Agency, Vienna, Austria) certified reference materials sucrose (IAEA-C-6; $\delta^{13}C$ = -10.8 ± 0.5‰; mean ± SD), ammonium sulfate (IAEA-N-2; $\delta^{15}N$ = 20.3 ± 0.2‰; mean ± SD) and silver sulfide (IAEA-S-1; $\delta^{34}S$ = -0.3‰) were used as primary analytical standards. Sulfanilic acid (Sigma-Aldrich; $\delta^{13}C$ = -25.6 ± 0.4‰; $\delta^{15}N$ = -0.13 ± 0.4‰; $\delta^{34}S$ = 5.9 ± 0.5‰; means ± SD) was used as a secondary analytical standard. Standard deviations on multi-batch replicate measurements of secondary and internal lab standards (coastal Mediterranean sediments) analysed interspersed with samples (one replicate of each standard every 15 analyses) were 0.2‰ for both $\delta^{13}C$ and $\delta^{15}N$ and 0.5‰ for $\delta^{34}S$.

**Benthic foraminiferal faunal abundance and diversity.**   The eight benthic foraminiferal samples per season were preserved in 4% formalin with phloxine B to stain the endoplasm of live benthic foraminifera [33]. The samples were wet sieved through 20, 100 and 1000 μm mesh sizes. Only the fraction > 100 μm was used for this study. Samples were sorted by hand under a stereo microscope and stained benthic foraminifera were collected and preserved on micropalaeontological slides. The faunal densities for each layer were standardised for a sediment volume of 35 cm$^3$. The species richness (S), corresponding to the observed number of species present in a sample, Shannon index H′ [38] for species diversity, and the evenness E as eH′/S of Buzas and Gibson (1969) [39] were used to compare the diversity of species between samples. PAST software was used to calculate these different indices [40]. Rarefaction curves were also drawn for each station to evaluate the completeness of our sampling approach. Species representing less than 5% were grouped together in the 'other' category.

**Foraminifera fluorescence.**   *Haynesina germanica* and *Quinqueloculina seminula* from all samples, previously manually isolated and sorted by species on microslides (plummer cell) were imaged. A motorised stereo Zeiss AxioZoom V16 microscope equiped with a HXP-120 light source and a Plan NeoFluar 1.0X was used at 20X total magnification (pixel size = 5.16 μm). Brightfield and red fluorescence channels (Ex 559–585 nm, Em 600–690 nm, BeamSplitter 590 nm) were imaged with an AxioCam HR R3. The red fluorescence channel

covers the chlorophyll autofluorescence spectra. Images were processed on Fiji software, an open-source platform for biological-image analysis [41], to measure the sum of the value of pixels of the fluorescence channel and the relative fluorescence area of individual *H. germanica* and *Q. seminula*. Briefly, an image mask was automatically generated by Huang threshold on the bright field channel to isolate each specimen on one ROI (region of interest). The threshold method proposed by Otsu et al. in 1979 [42] was used in each previous ROI on the fluorescent channel before measurement of fluorescence intensity as RawIntDen (the sum of all pixel values in the ROI) and % area.

**Statistical analyses.** Environmental variables were first treated by principal components analysis (PCA) to visualize their spatio-temporal distribution. In order to characterise the distribution of the most dominant species in the area in relation to environmental conditions, a redundancy analysis (RDA) was performed, and an ANOVA then used to evaluate the significance of the RDA results [43]. After selecting the environmental variables (temperature, TOC, TC, grain size, $\delta^{34}$S and oxygen) for the RDA, the adjusted $R^2$ value was calculated to discriminate the environmental variables that explained most of the variance in species abundance. The ANOVA produced a p-value for the model equal to 0.001, confirming that it is statistically significant.

## Results

### a) Inter-seasonal variability

**Environmental context.** Most environmental parameters showed little or no seasonal variation, except in August, when temperatures became warmer and oxygen more depleted: up to 11°C and 132 μM, respectively (Table 1). The pH showed local variations, such as strong increases in May and August at stations 2, 3 4, 7 and 8 and in August only at station 6. Salinity only showed slight variations apart from an abrupt decrease at station 5 in May and August. Analysis of SOMLIT seawater environmental data in the channel revealed clearer trends across seasons (S1 Fig). The TOC was stable over the four seasons with a notable exception at station 6 where there was a significant decrease of 0.7% in May. Similarly, TC was stable between seasons, except in February at station 7 where there was a significant increase to 2.3% (Table 1). In May, salinity decreased while oxygen, pH, and Chl *a* concentration increased drastically. Interestingly, such an increase was not found in the HPLC analyses of environmental Chl *a* or phaeopigment.

To corroborate the presence of a phytoplankton efflorescence, two species of foraminifera from different seasons were imaged in the red fluorescence channel for chlorophyll and organic matter content. The primary aim was to investigate the kleptoplasticity potential of the species *H. germanica* compared with the non-kleptoplastic species *Q. seminula*. Images of fluorescence of the two species are shown in S2 Fig. Fluorescence was observed on the two species of foraminifera throughout the seasons studied (Fig 2). Fluorescence intensity in August was very low for both species (Fig 2C and 2D). A very high fluorescence intensity was, however, observed in May for *H. germanica*, which was higher than at the other seasons (Fig 2C). However, the area of fluorescence per specimen in May did not increase significantly compared with the intensity of fluorescence over this period (Fig 2A and 2B).

**Faunal signal.** The overall abundances per season were low in May with a mean of 262 individuals/100 cm$^2$ per station. The density was about 10 times lower at this time than the greatest abundance of 2962 ind./100 cm$^2$ found in August (Fig 3).

Overall, a total of 29 species were identified, of which the nine that represented more than 5% (Fig 3) were imaged using a scanning electron microscope (Fig 4). The species diversity was much higher in August, with 24 species identified, compared with lower counts of 10 in

**Table 1. Environmental parameters characterising the eight sampling stations during the four seasons of the study in the Bay of Roscoff (2021–2022).**

| Sample station | Longitude | Latitude | Sampling date (dd/mm/yyyy) | Temperature (°C) | Oxygen concentration (µM) | pH | Salinity | Grain size (µm) $Q_{50}$ | total carbon (TC) (wt,%) | Total Organic Carbon (TOC) (wt,%) | Azote (%N) | Souffre (%S) | $\delta^{13}C$ (‰) | $\delta^{15}N$ (‰) | $\delta^{34}S$ (‰) | Chlorophylle $a$ (µg de chl$a$/cm$^2$) | Phaeopigment (Phaeo/chl$a$) |
|---|---|---|---|---|---|---|---|---|---|---|---|---|---|---|---|---|---|
| 1 | -4,00083 | 48,72318 | 09/11/2021 | 15,1 | 51,5 | 7,7 | 33,8 | 276 | 0,5 | 0,1 | 0,03 | 0,02 | -19,2 | 7,1 | 11,6 | 27,4 | 0,0 |
| 2 | -3,99956 | 48,72213 | 09/11/2021 | 15,1 | 153,7 | 7,7 | 36,0 | 168 | 1,5 | 0,1 | 0,03 | 0,02 | -20,4 | 7,8 | 12,2 | 25,0 | 0,1 |
| 3 | -3,99738 | 48,72146 | 09/11/2021 | 16,2 | 369,8 | 8,0 | 14,9 | 250 | 0,4 | 0,1 | 0,02 | 0,01 | -21,3 | 9,9 | 13,9 | 13,9 | 0,3 |
| 4 | -3,99325 | 48,72076 | 09/11/2021 | 16,9 | 250,0 | 7,8 | 33,6 | 226 | 0,3 | 0,1 | 0,03 | 0,01 | -19,9 | 5,9 | 12,9 | 33,5 | 0,1 |
| 5 | -4,00066 | 48,71257 | 09/11/2021 | 15,8 | 60,9 | 7,9 | 34,2 | 276 | 0,4 | 0,2 | 0,04 | 0,02 | -21,9 | 7,5 | 6,2 | 26,6 | 0,0 |
| 6 | -3,98214 | 48,72569 | 09/11/2021 | 15,4 | 8,3 | 7,8 | 34,1 | 125 | 1,7 | 1,2 | 0,13 | 0,15 | -21,6 | 6,1 | -0,8 | 18,1 | 0,3 |
| 7 | -3,98521 | 48,73121 | 09/11/2021 | 15,6 | 331,9 | 7,5 | 32,5 | 336 | 0,8 | 0,1 | 0,03 | 0,02 | -20,0 | 7,6 | 8,6 | 33,1 | 0,0 |
| 8 | -3,98407 | 48,73162 | 09/11/2021 | 15,2 | 139,9 | 7,4 | 34,0 | 906 | 2,5 | 0,4 | 0,07 | 0,04 | -20,9 | 7,6 | 10,1 | 43,4 | 0,0 |
| 1 | -4,00083 | 48,72318 | 07/02/2022 | 12,7 | 71,3 | 7,6 | 34,4 | 276 | 0,5 | 0,1 | 0,03 | 0,04 | -18,4 | 7,3 | 10,5 | 39,8 | 0,1 |
| 2 | -3,99956 | 48,72213 | 07/02/2022 | 12,2 | 0,9 | 7,8 | 29,8 | 168 | 1,6 | 0,2 | 0,03 | 0,05 | -21,4 | 7,6 | 10,6 | 19,5 | 0,1 |
| 3 | -3,99738 | 48,72146 | 07/02/2022 | 15,0 | 261,3 | 7,2 | 26,5 | 336 | 0,4 | 0,1 | 0,04 | 0,04 | -20,1 | 10,0 | 12,8 | 60,2 | 0,0 |
| 4 | -3,99325 | 48,72076 | 07/02/2022 | 13,5 | 28,2 | 7,6 | 32,9 | 250 | 0,3 | 0,1 | 0,03 | 0,04 | -20,6 | 7,2 | 9,1 | 30,4 | 0,1 |
| 5 | -4,00066 | 48,71257 | 07/02/2022 | 12,9 | 1,4 | 7,7 | 33,6 | 276 | 0,3 | 0,2 | 0,04 | 0,05 | -21,0 | 7,7 | 3,9 | 32,1 | 0,0 |
| 6 | -3,98214 | 48,72569 | 07/02/2022 | 12,4 | 31,0 | 7,7 | 31,1 | 113 | 1,5 | 1,1 | 0,07 | 0,20 | -21,1 | 6,2 | -6,9 | 5,7 | 0,4 |
| 7 | -3,98521 | 48,73121 | 07/02/2022 | 11,9 | 125,4 | 7,4 | 36,7 | 371 | 2,3 | 0,1 | 0,11 | 0,21 | -20,4 | 6,5 | -3,4 | 49,2 | 0,1 |
| 8 | -3,98407 | 48,73162 | 07/02/2022 | 11,7 | 118,9 | 7,6 | 32,2 | 250 | 2,4 | 0,3 | 0,05 | 0,06 | -20,7 | 7,5 | 9,8 | 18,3 | 0,1 |
| 1 | -4,00083 | 48,72318 | 03/05/2022 | 14,9 | 63,6 | 7,9 | 37,8 | 276 | 0,4 | 0,1 | 0,02 | 0,04 | -19,6 | 7,8 | 8,6 | 19,2 | 0,0 |
| 2 | -3,99956 | 48,72213 | 03/05/2022 | 16,2 | 218,0 | 8,1 | 33,5 | 168 | 1,2 | 0,2 | 0,03 | 0,05 | -20,6 | 7,9 | 12,0 | 22,3 | 0,0 |
| 3 | -3,99738 | 48,72146 | 03/05/2022 | 16,3 | 307,9 | 8,4 | 31,2 | 305 | 0,4 | 0,2 | 0,04 | 0,05 | -19,8 | 10,0 | 10,5 | 76,5 | 0,0 |
| 4 | -3,99325 | 48,72076 | 03/05/2022 | 16,7 | 350,3 | 8,8 | 34,6 | 226 | 0,2 | 0,1 | 0,02 | 0,04 | -20,6 | 7,5 | 9,4 | 25,8 | 0,1 |
| 5 | -4,00066 | 48,71257 | 03/05/2022 | 14,3 | 2,6 | 8,0 | 15,6 | 276 | 0,4 | 0,3 | 0,04 | 0,07 | -21,4 | 7,4 | -0,5 | 45,3 | 0,1 |

*(Continued)*

**Table 1.** (Continued)

| Sample station | Longitude | Latitude | Sampling date (dd/mm/yyyy) | Temperature (°C) | Oxygen concentration (μM) | pH | Salinity | Grain size (μm) $Q_{50}$ | total carbon (TC) (wt,%) | Total Organic Carbon (TOC) (wt,%) | Azote (%N) | Souffre (%S) | $\delta^{13}C$ (‰) | $\delta^{15}N$ (‰) | $\delta^{34}S$ (‰) | Chlorophylle a (μg de chla/cm²) | Phaeopigment (Phaeo/chla) |
|---|---|---|---|---|---|---|---|---|---|---|---|---|---|---|---|---|---|
| 6 | -3,98214 | 48,72569 | 03/05/2022 | 15,5 | 44,0 | 7,8 | 36,0 | 125 | 1,1 | 0,7 | 0,07 | 0,15 | -21,6 | 6,1 | -7,7 | 18,4 | 0,1 |
| 7 | -3,98521 | 48,73121 | 03/05/2022 | 13,9 | 25,9 | 7,6 | 36,3 | 371 | 0,7 | 0,1 | 0,03 | 0,04 | -20,7 | 7,5 | 11,5 | 35,5 | 0,0 |
| 8 | -3,98407 | 48,73162 | 03/05/2022 | 14,8 | 105,7 | 7,9 | 35,8 | 673 | 2,3 | 0,3 | 0,06 | 0,06 | -20,6 | 7,4 | 9,2 | 48,6 | 0,0 |
| 1 | -4,00083 | 48,72318 | 11/08/2022 | 20,8 | 10,4 | 7,4 | 35,8 | 276 | 0,5 | 0,1 | 0,03 | 0,05 | -17,9 | 6,4 | 9,1 | 51,5 | 0,1 |
| 2 | -3,99956 | 48,72213 | 11/08/2022 | 22,9 | 22,7 | 8,0 | 35,8 | 186 | 1,3 | 0,2 | 0,03 | 0,06 | -21,0 | 7,5 | 9,8 | 23,6 | 0,0 |
| 3 | -3,99738 | 48,72146 | 11/08/2022 | 24,3 | 116,6 | 8,5 | 31,8 | 276 | 0,5 | 0,2 | 0,03 | 0,05 | -18,5 | 9,7 | 9,2 | 41,5 | 0,0 |
| 4 | -3,99325 | 48,72076 | 11/08/2022 | 24,8 | 126,1 | 8,4 | 34,5 | 250 | 0,2 | 0,1 | 0,02 | 0,04 | -20,7 | 7,6 | 8,8 | 13,3 | 0,1 |
| 5 | -4,00066 | 48,71257 | 11/08/2022 | 23,9 | 3,8 | 7,6 | 10,3 | 250 | 0,4 | 0,3 | 0,04 | 0,08 | -20,5 | 7,8 | 0,9 | 20,8 | 0,1 |
| 6 | -3,98214 | 48,72569 | 11/08/2022 | 28,2 | 1,0 | 8,4 | 33,0 | 102 | 1,8 | 1,4 | 0,13 | 0,28 | -21,0 | 6,3 | -9,6 | 20,8 | 0,0 |
| 7 | -3,98521 | 48,73121 | 11/08/2022 | 21,7 | 3,2 | 7,5 | 34,8 | 371 | 0,5 | 0,1 | 0,02 | 0,04 | -20,6 | 7,7 | 8,2 | 32,8 | 0,0 |
| 8 | -3,98407 | 48,73162 | 11/08/2022 | 24,1 | 19,5 | 7,8 | 34,8 | 673 | 1,7 | 0,2 | 0,04 | 0,06 | -20,7 | 7,5 | 9,5 | 30,5 | 0,0 |

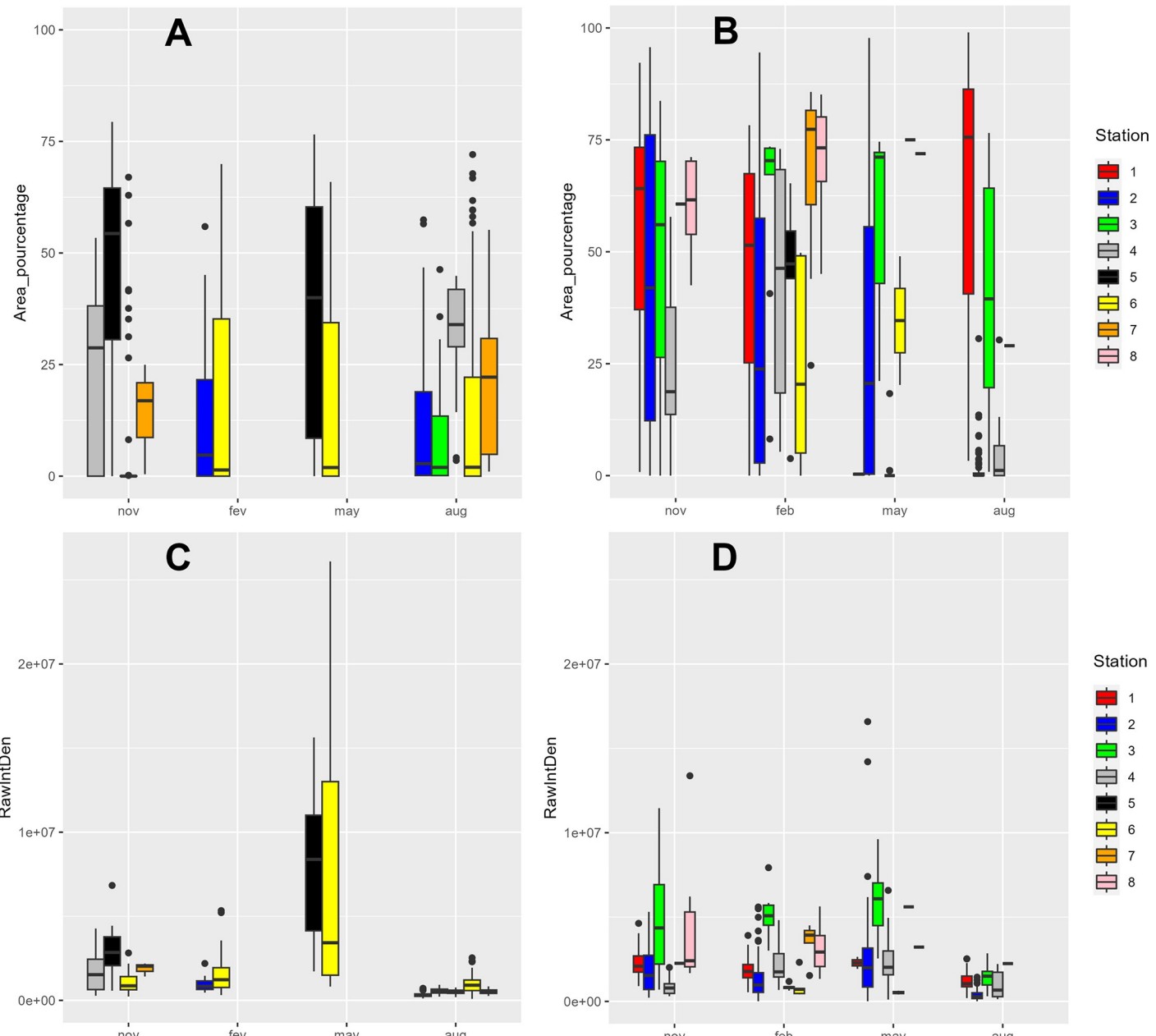

**Fig 2. Percentage of fluorescence area.** (A) *Haynesina germanica* and (B) *Quinqueloculina seminula*; fluorescence intensity for (C) *Haynesina germanica* and (D) *Quinqueloculina seminula*.

May, and 13 in February and November. However, the mean of the Shannon index at each station was similar between August and February (~ 1.15) and was much lower in May (0.77). The Evenness index values remained quite stable (0.60–0.77) except for a marked decrease in August (0.41). The rarefaction curves indicate that the sampling effort was generally sufficient to cover all species diversity, apart from in May (Fig 5).

Among all the species found, the most dominant was *Q. seminula*, with 6833, 4428 and 1188 ind./100 cm$^2$ in November, February and May, respectively. High abundances were also observed for *Cribroelphidium gerthi* in November (3197 ind./100 cm$^2$), and *Elphidium crispum*

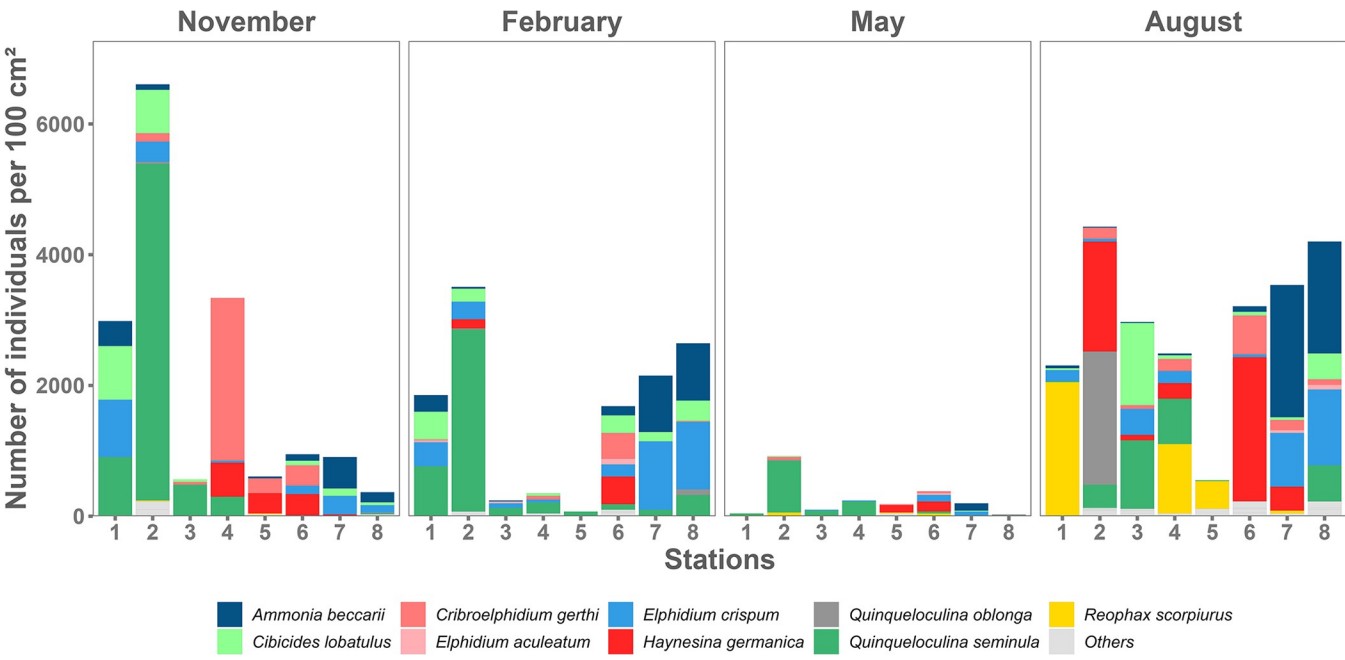

**Fig 3. Densities (expressed in number of specimens per 100 cm²) of the major (>5%) living species of benthic foraminifera in the Bay of Roscoff at the eight sampling stations for the four months of the study.**

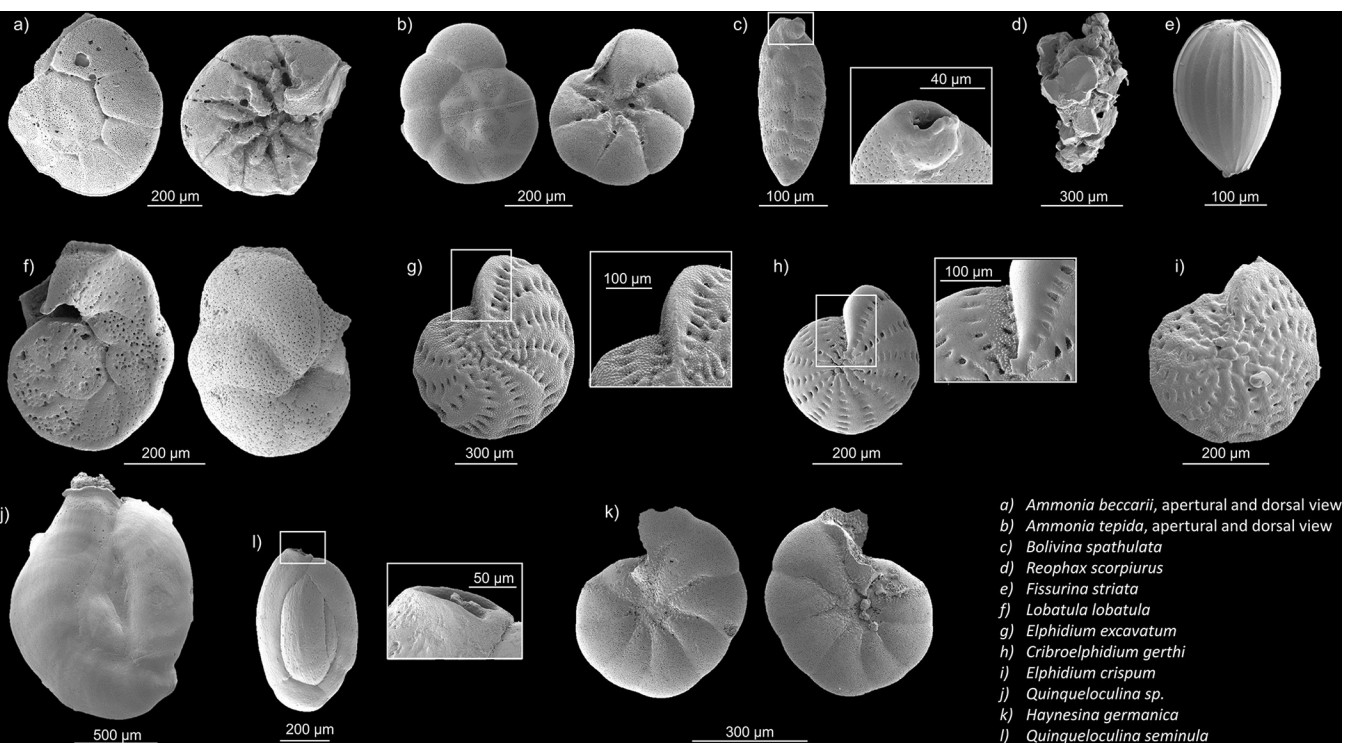

**Fig 4. SEM images of the species of foraminifera found in the various stations sampled in Roscoff Aber Bay.**

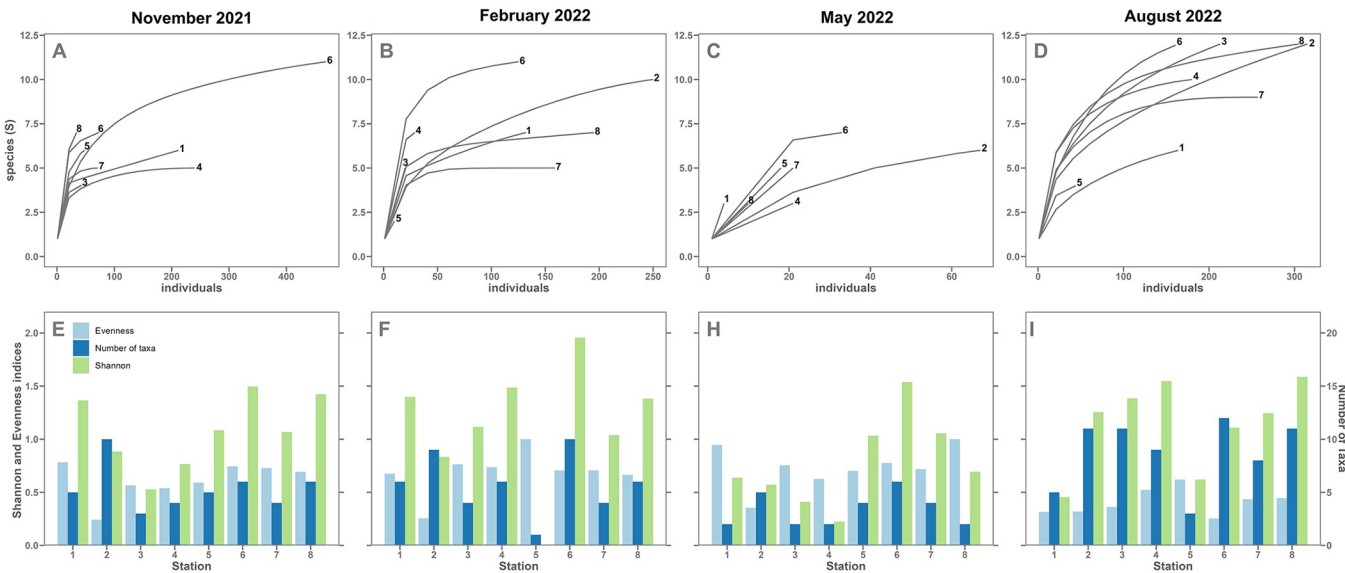

**Fig 5.** A) to D) Rarefaction curves for all stations from November 2021 to August 2022, E) to I) Shannon index H' (green), Evenness index E (light blue) and number of taxa (dark blue) based on observed living individuals.

in February (3013 ind./100 cm$^2$). The highest species dominance occurred in August, as reflected by the Evenness index, with *H. germanica* (4584 ind./100 cm$^2$), *Ammonia beccarii* (3919 ind./100 cm$^2$) and *Reophax scorpiurus* (3579 ind./100 cm$^2$) showing particularly high occurrences (Fig 3).

## b) Intra-seasonal variability

**Spatial characterisation.** All stations had fine to medium sandy particles (168–371 μm), except for station 8, where they were coarser (~900 μm). Station 6 had the finest sediments, varying between 102 and 125 μm. The salinity was homogeneous among almost all the sites except the estuarine station 5 and station 3 (down to 10–15) influenced by terrestrial freshwater (Table 1). The pH values were higher in the bay and harbour stations (~7.8–8.2) than in the channel stations (~7.6). Temperature was also higher in the bay and harbour stations (Table 1). The TOC values were all approximately similar (0.10–0.31) except for station 6 in the harbour (1.09), which was up to ~900% higher. The opposite trend was observed regarding oxygen availability, with minima observed at the harbour station 6 (21 μM) and estuarian station 5 (17 μM) and maxima at bay stations 3 and 4 (188–264 μM). The harbour station 6 also presented a remarkably higher sulfur percentage (0.20%) and thus lower δ$^{34}$S values (-6.23‰) compared with the other stations (0.03–0.08% and 2.61–11.63‰). Similarly, station 6 had the lowest Chl *a* (15.75 μg Chl *a*/cm$^2$) and highest phaeopigment concentrations (0.22 Phaeo/Chl *a*, Table 1).

**Species distribution.** Total density was similar among all stations (~6000–7000 ind./100 cm$^2$) except for a very high density at station 2 (15463 ind./100 cm$^2$) and very low values at Stations 3 and 5 in the bay (Fig 3).

*Quinqueloculina seminula* was the most abundant species inside the bay (stations 1, 2, 3 and 4) reaching up to 59% at station 2. The other main species in the bay were *R. scorpiurus*, 17–32% at stations 1, 4 and 5, *H. germanica*, 12–30% at stations 2, 4 and 5, *C. lobatulus*, 6–33% at stations 1–3 and *E. crispum*, 4–20% at stations 1–4. Channel stations 7 and 8 were characterised by the dominance of *Ammonia beccarii* (38–51%) and *E. crispum* (31–32%). Harbour station 6 was mainly dominated by *H. germanica* (50%), followed by *C. gerthi* (21%).

## Discussion

### a) Inter-seasonal variability

**Seasonal climatic events.**   Seasonal variations had a major influence on the distribution, abundance and diversity of benthic foraminiferal species. Samples from the month of May provide a striking example, with a drastic decline in the abundance and diversity of foraminiferal populations. Environmental parameters recorded at each station failed to provide a clear explanation for this reduction of abundance since abiotic factors and food avaibility proxies (i.e. TOC and Chl *a*) remained stable, suggesting that a disturbance may have occurred before sampling. The SOMLIT data from the same period reveals the presence of a spring phytoplanktonic bloom in the channel, likely resulting from terrestrial nutrients washed in by precipitation (S3 Fig). Previous studies have described phytoplanktonic diversity at the SOMLIT-Astan site and revealed a dominance of diatoms in May [23]. A high increase of red fluorescence intensity at this period in *H. germanica*, certainly related to the presence of chlorophyll inside the cells (Fig 2), could possibly be interpreted as a kleptoplasty signal, as *H. germanica* is known for maintaining living diatoms [44, 45]. However, the same pattern of fluorescence intensity was also observed in the non-kleptoplastidic species *Q. seminula*. Thus, the kleptoplasty signal in *H. germanica* cannot be attributed to feeding on phytobenthos bloom by our method. Further in situ investigation would be necessary to explore kleptoplasty activity of *H. germanica* that could explain the success of this species in impacted systems. The most probable hypothesis would thus be that climatic events (i.e. storms with strong wind gusts) have periodically disturbed the ecosystem, resulting in the washing or mixing of sediments [46]. The data obtained in May would therefore reflect an intermediate phase following significant local ecosystem disturbances but preceding the end of the bloom, which could introduce new organic matter inducing recovery of a stable state (S1 Fig). This decrease of faunal abundance could also be explained by amensalism or competition between foraminifera and other meiofaunal taxa (i.e. nematodes, copepods, [47]). Previous studies have also demonstrated a major impact of bacterial communities on foraminifera that might have contributed to population disturbance [46]. A final hypothesis could be that observed variations of abundance across seasons resulted from an annual life cycle [48]. However, this seems unlikely considering that foraminiferal species reproduce on a continuous basis [49]. Overall, additional studies with finer temporal resolution would be necessary to test these hypotheses. These results suggest that monitoring of foraminiferal abundance and diversity could be a potential indicator of an extreme climatic event at a local scale.

**Seasonal species response.**   August was characterised by a very high species diversity and a high abundance of few species (Fig 3). *Quinqueloculina oblonga* and *R. scorpius*, which had low abundance in other seasons, were highly abundant at this time.

Abundance data suggests that *Q. oblonga* partially replaced *Q. seminula* due to a shift in ecological niche conditions more favourable to the former. One explanation could be that *Q. oblonga* may be able to withstand higher salinity [50]. This increase in salinity may be due to a higher evaporation during low tides resulting from higher temperatures in August.

*Reophax scorpiurus* was present in low abundance in May but became highly abundant in August. At the same period, the presence of agglutinants (*Reophax* genus), can be explained by a drastic temperature increase and eutrophied conditions. This species is known to tolerate a wide range of physico-chemical conditions [57]. The decline of the most abundant species, including *Q. seminula*, certainly due to the decrease in oxygen coupled with the drop in foraminiferal populations in May, opened up new ecological niches for *R. scorpiurus*. This species could represent a pioneer colonizer following the event in May that affected the foraminiferal community as a whole. It has been observed in highly variable trophic conditions and seems

able to tolerate variable OM quality [51, 52]. We can postulate that this species is more competitive when the organic matter present is more refractory and after an environmental disturbance, hence representing a potential bio-indicator for putative stresses.

## b) Intra-seasonal variability

The stations were chosen to provide a range of environmental conditions within a small area, considering both natural and anthropogenic influences. These included freshwater inputs, harbour conditions, tide levels and channel influence. A PCA was used to characterise the different study sites by correlating stations with environmental parameters (Fig 6), providing a comprehensive spatial repartition of the foraminiferal species. The first axis of the PCA explains 34.85% of the variability, with nitrogen, sulfur, TC, phaeopigment and TOC parameters positively loaded (corresponding to eutrophic environments) and oxygen, $\delta^{34}S$, $\delta^{15}N$ and $\delta^{13}C$, negatively loaded (corresponding to oligotrophic environments). Station 6 (harbour)

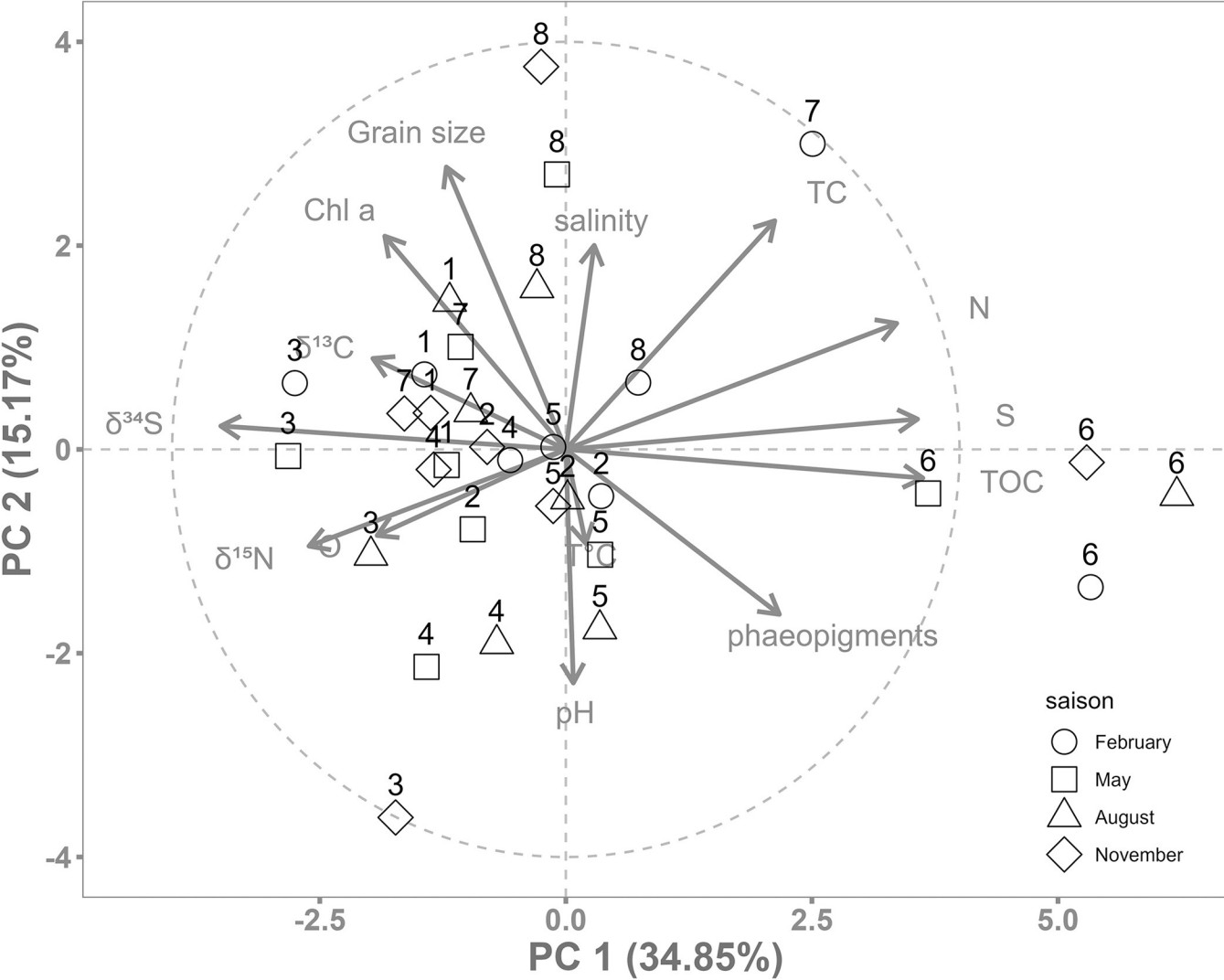

**Fig 6. Principal component analysis based on environmental parameters.** Grain size, Oxygen (O), pH, Temperature (T˚C), salinity, Chlorophyll *a* (Chl *a*), phaeopigments, Total organic carbon (TOC), Total Carbon (TC), Nitrogen (N), Sulfur (S), $\delta^{13}C$, $\delta^{15}N$, $\delta^{34}S$ sampled during four seasons at the eight sampling stations.

was strongly correlated with parameters that are indicators of eutrophic environments, which is consistent with the influence of fishing waste being discharged into this area, resulting in a high concentration of organic matter (TOC, phaeopigments, [21]). The phaeopigments were higher in the eutrophic zone due to the degradation of Chl *a* in the harbour and terrestrial zones (Aber Bay). Also, δ$^{34}$S and δ$^{13}$C were lower in the eutrophic domain, reflecting a higher isotope fragmentation of sulfur and carbon in the harbour probably caused by high biological activity (Table 1). This eutrophic domain (station 6) was dominated by *H. germanica* and *C. gerthi*, the latter being able to live in low-oxygen environments [53]. By combining the high TOC values from station 6 and the abundance of *H. germanica* according to the Foram-AMBI ecological index, this species could be classified in group III—IV, which includes the '*Tolerant species*' and '*2nd-order opportunistic species*', as this taxon increases significantly towards sites of maximum organic enrichment [12]. *Haynesina germanica* has already been identified as a bio-indicator of pollution correlated with anthropic activities, such as pollutants introduced by industrial activity [54, 55]. Conversely, the bay stations 1 to 4 represented the oligotrophic domain. PC2 explains 15.17% of the environmental variability (Fig 6) with grain size, salinity, Chl *a* and TC positively loaded and pH and temperature negatively loaded. Channel stations 7 and 8 are strongly driven by positive parameter of PC2, highlighting the dynamic context of this nearly open water system. This more dynamic subtidal zone is characterised by a dominance of *A. beccarii* and *E. crispium* (Fig 7), which both prefer more marine conditions and thus benefit from the channel influence [56]. *Ammonia beccarii* has been shown to be highly tolerant to variations in substratum types and total organic carbon [57]. Conversely, Aber bay stations 2 to 5 are associated with terrestrial influences, while stations 3 and 5 were particularly influenced by freshwater inflows [58]. Deposits of green algae due to the ephemeral proliferation of *Enteromorpha* spp. are also present at these stations [59], linked to major inputs of nutrients [60]. Another freshwater outlet is also present at station 4, which is situated close to a thalassotherapy centre. In this oligotrophic domain, *Q. seminula* is dominant. Higher oxygen concentrations were observed at station 2 (Fig 7), confirming that *Q. seminula* thrives in well-oxygenated environments [61, 62]. This species is also known to be associated with areas with high hydrodynamics and has a preference for areas with marine channel influence [56, 63]. It can also feed on phytodetritus and is probably capable of rapidly ingesting fresh organic matter from marine primary production [64, 65].

**Bio-indicator species.** Across seasonal and spatial responses of foraminiferal communities, some species appear to be good describers of their environment and/or perturbations that occurred. We performed an RDA analysis to evaluate the major species response to environmental conditions, in order to define possible bio-indicator species (Fig 7). *Cribroelphidium gerthi* appeared to be positively correlated with the eutrophic domain together with *H. germanica*, a species already used as bio-indicator of pollution from anthropogenic activities that might thus be in turn an interesting species to monitor human impact, as previously suggested by [66]. The species *Reophax scorpius* was markedly associated with the increase of temperature and would thus represent a good indicator to monitor large environmental disturbances induced by global warming at a local scale. Considering the burden of evaluating global warming, we can put forward this species as a potential bio-indicator candidate of increasing temperature over time.

We also observed that *Bolivina* sp. showed calcareous shell deformations, consisting in a double aperture, at station 6 during the August sampling (S4 Fig). Coastal environments characterised by anthropogenic activities often suffer from increased pollution, including heavy metals [2], that negatively impact the abundance and diversity of foraminifera [67], as well as the development of calcareous shell deformations [68]. In this study, even though heavy metals were not investigated, the presence of this abnormality suggests a potential enrichment of such

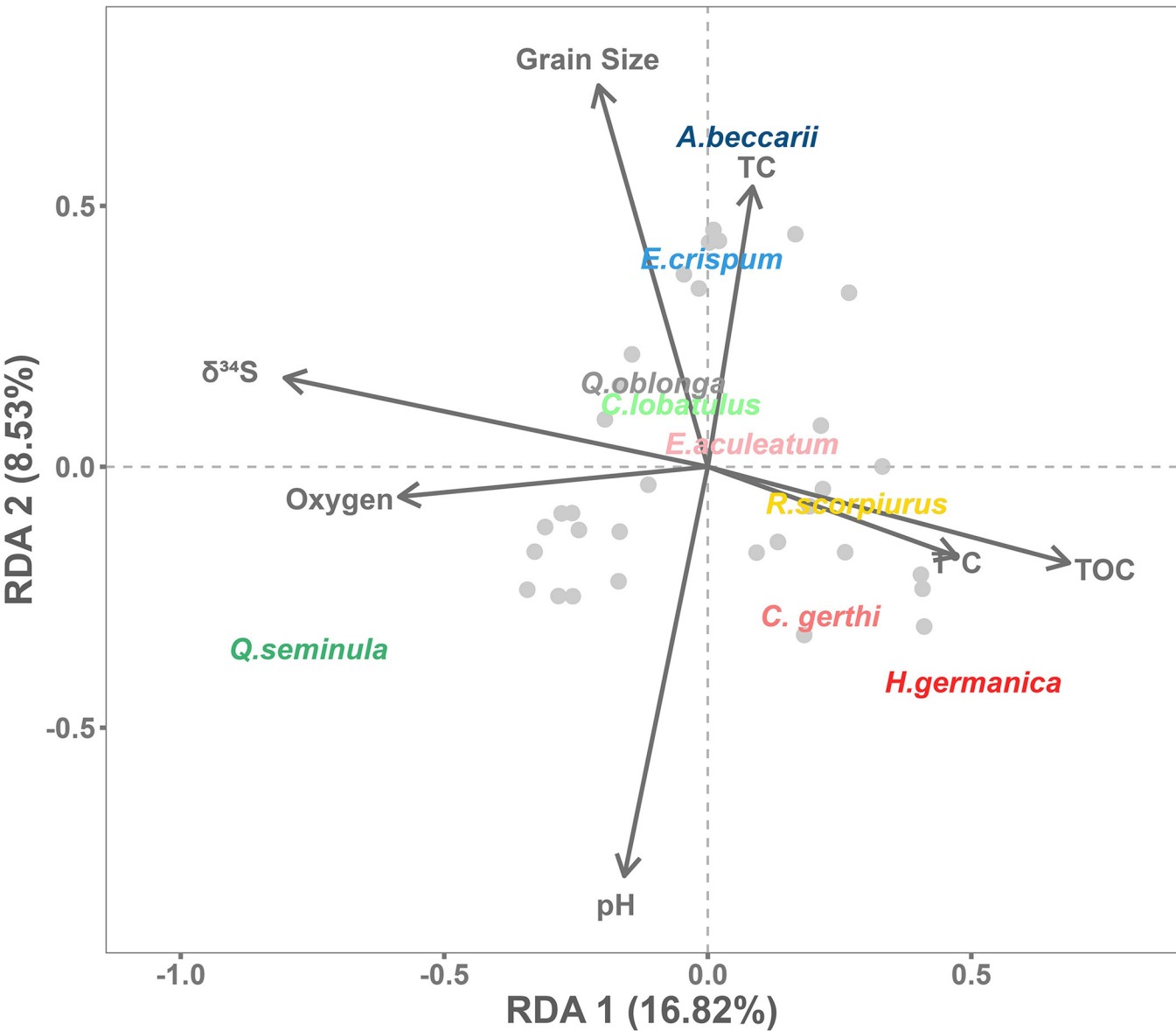

**Fig 7. Redundancy analysis (RDA) of foraminiferal species with environmental variables.** The species are *Ammonia beccarii*, *Elphidium crispum*, *Elphidium aculeatum*, *Cibicides lobatulus*, *Cribroelphidium gerthi*, *Haynesina germanica*, *Quinqueloculina seminula*, *Quinqueloculina oblonga*, *Reophax scorpiurus* (Hellinger-transformed abundance) and the environmental variables are Grain Size, Total Carbon (TC), Temperature (T°C), Total Organic Carbon (TOC), Oxygen (O), δ³⁴S, pH, P < 0.05.

contaminants. However, the low occurrences of these abnormalities in foraminiferal individuals indicates a non-significant impact of contaminants in the area. Furthermore, the harbour station hosts a quite diverse Foraminifera community overall, suggesting that the eutrophication from fishing waste does not compromise biodiversity and that an impact of pollution on Foraminifera in the area is unlikely. In this study we also observed that Foraminifera are influenced by natural stresses in estuaries, with a community response similar to anthropogenic stress. Other studies, such as Alve et al. in 2016 [10], showed even higher values of organic matter for this area, confirming that it is enriched, but not excessively so. It is worth noting that global diversity and abundance of the overall community can thus provide

complementary information to the presence of a bioindicator species. Indeed, in estuaries, the characteristics of natural stress may be similar to those of anthropogenic stress. This result confirms the 'estuarine paradox' and indicates that environmental indicators should be used with caution [16, 56, 69].

## Conclusion

Seasonal and spatial distribution of foraminiferal species in Roscoff Aber Bay reveal a very fine scale of environmental variability as well as significant disturbance events. Very clear distinct populations were identified in association with open water (*A. beccarii* and *E. crispum*), the oligothrophic inner bay (*Q. seminula*) and euthrophic human altered harbour (*H. germanica* and *C. gerthi*) (Fig 8). Among these species, only *H. germanica* has been identified in the literature as a bio-indicator of anthropogenic impacts, but the other species identified in this study are useful for temporal monitoring. Some species showed opportunistic behaviour, with sudden

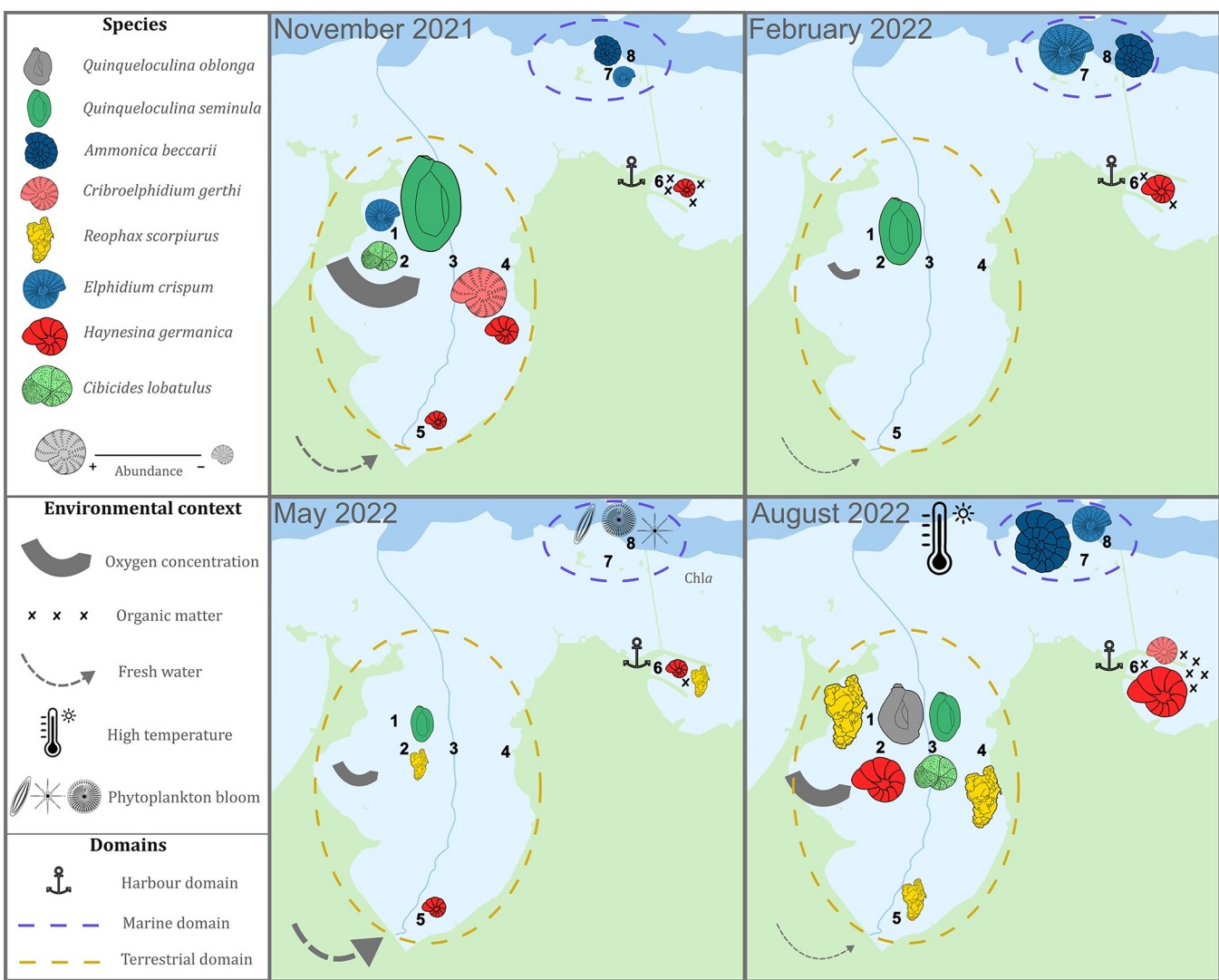

**Fig 8. Synthetic schema of major species distribution and abundance in Roscoff Aber Bay over four seasons with environmental variation.** The species are *Ammonia beccarii, Elphidium crispum, Cibicides lobatulus, Cribroelphidium gerthi, Haynesina germanica, Quinqueloculina seminula, Quinqueloculina oblonga, Reophax scorpiurus* and the environmental variables are oxygen concentration, organic matters and fresh water inputs.

increases after environmental disturbances and shifts in ecological niches (*R. scorpiurus* and *Q. oblonga*). The harbour station showed an environment enriched in organic carbon exceeding the natural background measured in the other stations. Foraminiferal diversity was not impacted by this pollution, except for the presence of species adapted to organic matter enrichment. This study confirmed the use of benthic foraminifera as indicators for characterising ecosystem health and demonstrated their use for surveying environmental dynamics, such as phytoplankton blooms, organic matter accumulation, and potential human-induced stress, at a seasonal scale. Within foraminifera, the fluorescence method has, with further investigation, the potential to characterise the onset of phytobenthos blooms and highlight potentially kleptoplastic species. With a view to future research work, this monitoring could help us to study the evolution of climatic conditions and anthropogenic activities in the harbour area to maintain the health of the current ecosystem.

## Supporting information

**S1 Fig. SOMLIT data for salinity, oxygen, pH and Chl *a* over the sampling period from November 2021 to August 2022 at the ESTACADE SOMLIT sampling point.** The data were extracted from the SOMLIT database (Service d'Observation en Milieu Littoral; www.somlit. fr) on 5 October 2022.
(TIF)

**S2 Fig. Plate of *Haneysina Germanica* foraminifera from station 4 in November 2022 in brightfield.** (A) and red fluorescence channels (Ex 559–585 nm, Em 600–690 nm, BeamSplitter 590 nm) (B) and an illustration of a density plot for red fluorescence (C).
(TIF)

**S3 Fig. Rainfall data for Morlaix (21 km from Roscoff Aber Bay in Brittany, France).** The period October 2021 to August 2022 from the infoclimat.fr/climatologie/globale/31-aout/morlaix/000AW.html website.
(TIF)

**S4 Fig. SEM image of *Bolivina* sp. in August 2022 at station 6 showing a morphological anomaly of a double aperture.**
(TIF)

## Acknowledgments

This work was supported by the BLUE REVOLUTION project (Biodiversity underestimation in our bLUe planEt: artificial intelligence REVOLUTION in benthic taxonomy) funded by the Interdisciplinary Graduate School for the Blue Planet (ISBlue; ANR-17-EURE-0015) and Ifremer (Institut français de recherche pour l'exploitation de la mer). DZ was supported by the project "Massive mEIOfauna DiscoverY of new Species of our oceans and SEAs (MEIODYS-SEA) funded by the Sasakawa Peace Foundation. RR was supported by the ISA-IFREMER Collaboration in support of the capacity development of national from developing States, by the Ifremer Marine Mineral Resources project (REMIMA project) and by the French National Research Agency under France 2030 (reference ANR-22-MAFM-0001). The authors thank Sarah Garric and Christophe Six for their help and support with the HPLC analysis of the sediment pigments and the interpretation of the various dendrograms. The authors would also like to thank Valentine Foulquier for her help with the field collection, extraction and technical analysis of the samples. The authors are grateful to the SOMLIT platform for providing temporal data from November 2021 to August 2022 for the ESTACADE point. The authors thank

Nicolas Gayet for his help in acquiring SEM images of the foraminifera. The authors would also like to thank Victor Simon for the loan of temperature, pH and salinity probes and Christelle Simon Colin for the loan of the freeze-drying machine. Finally, the authors would like to thank the scientific managers of the ESTACADE stations and the SOMLIT coordinator.

The authors declare that they have no conflicts of interests.

## Author Contributions

**Conceptualization:** Edwin Daché.

**Data curation:** Edwin Daché.

**Formal analysis:** Edwin Daché, Pierre-Antoine Dessandier, Ranju Radhakrishnan, Valentin Foulon, Loïc Michel.

**Investigation:** Edwin Daché, Pierre-Antoine Dessandier.

**Methodology:** Edwin Daché, Pierre-Antoine Dessandier, Valentin Foulon, Loïc Michel.

**Project administration:** Colomban de Vargas, Daniela Zeppilli.

**Supervision:** Pierre-Antoine Dessandier, Colomban de Vargas, Jozée Sarrazin, Daniela Zeppilli.

**Validation:** Pierre-Antoine Dessandier.

**Writing – original draft:** Edwin Daché.

**Writing – review & editing:** Pierre-Antoine Dessandier, Valentin Foulon, Loïc Michel, Jozée Sarrazin, Daniela Zeppilli.

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
