## [Decision Letter · Decision Letter 0]

23 Apr 2024

PONE-D-24-06208Benthic Foraminifera as bio-indicators of natural and anthropogenic stressesPLOS ONE

Dear Dr. Dache,

Thank you for submitting your manuscript to PLOS ONE. After careful consideration, we feel that it has merit but does not fully meet PLOS ONE’s publication criteria as it currently stands. Therefore, we invite you to submit a revised version of the manuscript that addresses the points raised during the review process.

We look forward to receiving your revised manuscript.

Kind regards,

Marcos Rubal García, PhD

Academic Editor

PLOS ONE

Journal Requirements:

3. Thank you for stating the following in your Competing Interests section: "No competing interests"

5. Please include your tables as part of your main manuscript and remove the individual files. Please note that supplementary tables (should remain/ be uploaded) as separate ""supporting information"" files

**Additional Editor Comments:**

Dear Authors, two external reviewers have now assessed your manuscript I have received the reports from referees on your manuscript, "Benthic Foraminifera as bio-indicators of natural and anthropogenic stresses", providing the comments that are reported below. As you can see, they both found your study needs important modifications. They identified a number of issues that would require careful revision before this paper is recommendable for acceptance.

Based on the reviewers' and my own assessment, I'm thus here inviting you to take all of these comments into careful consideration and to modify your manuscript according to the provided constructive suggestions. I will then be happy to receive and further examine your revised version together with a point-by-point reply to each comment by myself and each reviewer, where you will need to explain any changes done to a particular piece of text, or include supported and convincing counterarguments to any points you may disagree with I'm confident you will find the present comments and suggestions relevant and useful to improve your work and I'm thus looking forward to hearing back form you by the due time.

Reviewers' comments:

Reviewer's Responses to Questions

**Comments to the Author**

1. Is the manuscript technically sound, and do the data support the conclusions?

Reviewer #1: Yes

Reviewer #2: No

2. Has the statistical analysis been performed appropriately and rigorously? 

Reviewer #1: Yes

Reviewer #2: Yes

3. Have the authors made all data underlying the findings in their manuscript fully available?

Reviewer #1: Yes

Reviewer #2: Yes

4. Is the manuscript presented in an intelligible fashion and written in standard English?

Reviewer #1: No

Reviewer #2: No

5. Review Comments to the Author

Reviewer #1: See the attached Word copy of your ms. It is edited using Track Changes. The topic is well treated but the language needs some clarification. I did this for all sections except the Summary and Conclusions which needs to be rewritten to clarify and eliminate the wordiness.

Reviewer #2: Comments:

Dachè et al made a sincere effort to understand the response of benthic foraminifera in the coastal settings of Roscoff Aber Bay. The manuscript is poorly drafted and needs a significant improvement. I suggest authors to overhaul the result and discussion section. First start with inter-season, then move to intra-season details. Although, authors have generated a large number of dataset from the region but fails to provide a comprehensive finding and not able to marks the relationship between benthic foraminifera abundance and diversity with ambient ecological parameters.

My comments follow:

Line 21: remove “good”.

Line 79-81: ‘It is……….its salinity’. Provide references.

Line 103: remove ‘temporal’

Line 123: Mark the freshwater stream in figure 1 clearly.

Line 137-138: It is hard to distinguished that authors used only living benthic foraminifera or total benthic foraminifera population including dead shells (in the top 5 cm). Needs to mention clearly.

Line 157-160: Please provide accuracy of conductivity, pH and oxygen measurements.

Line 187: Sulfanilic acid was used as the elemental standard. Why acid was used as a standard for sediment samples, where standard and samples values deviate with a large difference.

Line 206-207: ‘4% formalin.....foraminiera’. cite reference.

Line 232: put space between % and area.

Line 252: ‘pH….sites. Remove the sentence. 7.2 to 8.8 is not a little variation by any mean.

Results: Results should be written first inter-season wise, followed by intra-seasonal variations. It is very hard to follow it in its current form.

Line 269: ‘Oxygen concentration was highest in November’? at which station? All of them? If yes, then mention. Also provide values for it rather than writing highest and lowest. Correct it throughout the result section.

Line 287: δ13C, δ15N, δ34S. Make mass value superscript.

Line 309-311: The δ34S………seasons. It does not belong to result section. Remove it.

Line 430: Micohabitats section fails to provide any details of benthic foraminifera microhabitat. It is just full of previously published information.

Line 541: H. germanica. Write it in Italics.

Line 542: Q. seminula. Write it in Italics.

Line 545: Q. seminula. Write it in Italics.

Line: Discussion section read redundant and need complete overhauling.

6. PLOS authors have the option to publish the peer review history of their article (what does this mean?). If published, this will include your full peer review and any attached files.

Reviewer #1: **Yes: **Jere H. Lipps

Reviewer #2: No

---

## [Author Response · Author response to Decision Letter 0]

25 Jul 2024

PLOS ONE Editorial Office

Object: « Benthic Foraminifera as bio-indicators of natural and anthropogenic conditions in Roscoff Aber Bay (Brittany, France)” by Edwin Daché and coauthors

Brest, 08/07/2024

Dear Editor,

We are grateful for the pertinent suggestions that you and the two reviewers have made, which have greatly contributed to the quality and clarity of this new version of our manuscript. 

We have carefully addressed all the issues and comments raised and we have highlighted each change in this point-by-point response, detailed below. The structure of the manuscript has been modified as requested by reviewers and a professional high-standard English language editing has been performed. 

I am therefore pleased to send you the modified version of the manuscript entitled "Benthic Foraminifera as bio-indicators of natural and anthropogenic conditions in Roscoff Aber Bay (Brittany, France)" by Edwin Daché and co-authors.

Sincerely yours,

Edwin Daché

Point to Point Response to reviewers’ comments

Reviewer reports

Reviewer 1:

Reviewer 1: See the attached Word copy of your ms. It is edited using Track Changes. 

AUTHORS: We thank the Rev1 for this improvement of the manuscript. All corrections proposed by reviewer 1 have been integrated into the text (please see below the detailed replies for proposed modifications).

Reviewer 1: The topic is well treated but the language needs some clarification. I did this for all sections except the Summary and Conclusions which needs to be rewritten to clarify and eliminate the wordiness.

AUTHORS: We agree with Rev1 that an English check was necessary as well as a rewriting of the Summary and Conclusion. The structure of the manuscript has been modified as requested by reviewers and a professional high-standard English language editing has been performed.

Reviewer 1: Line 2: change “stresses” by conditions

AUTHORS: Modified

Reviewer 1: Line 28: What do you mean? What are the characteristics of these forams?

AUTHORS: This sentence has been clarified and we modified the term “opportunistic species” by “sensitive species to particular conditions”. 

Lines 26-28:” We sought to understand the spatial distribution of foraminiferal populations within and between sampling sites over the different seasons and to identify sensitive species and those tolerant to anthropogenic impacts.”

Reviewer 1: Line 43: What kinds of activities do this?

AUTHORS: We thank Rev1 for this question. The text was not clear, in the amended version we added in the text clarifications on that: “anthropogenic activities (fisheries, trash deposits, etc.)”

Lines 43-45: “Interestingly, the observed organic enrichment of the harbour due to anthropogenic activities (fisheries, waste deposits, etc.) does not seem to significantly affect foraminiferal diversity.”

Reviewer 1: Lines 47-48: This is not true. Maybe for the meiobenthos but not the marine environment.

AUTHORS: We modified the sentence accordingly: 

Lines 49-50: “Foraminifera are a highly abundant phylum that dominate the meiobenthos as the most diverse group of shelled organisms [1].”

Reviewer 1: Line 70: What are these? Seasonal sampling ++

AUTHORS: We thank Rev1 for this question, however we are not sure about the sense of this request. We interpreted that the remark was about the sampling effort, so we added the following lines in the text.

Lines 70-74: “Intertidal areas are also characterised by strong seasonal variability that is often neglected due to the high sampling effort required for faunal and environmental analyses in each season. This underlines the necessity for additional studies in environmental monitoring, utilizing indices that incorporate comprehensive datasets and account for seasonal variations in these environments.”

Reviewer 1: Lines 89-90: I am not sure what is being described here. What is the “workshop zone”? Is it in the Station or the coast around or near the station and how is it delimited? Not clear.

AUTHORS: As requested by Rev1, we clarified this sentence. The foundation of the Biological Station in Roscoff was mainly due by the fact that the coastline surrounding Roscoff is a well-known hot-spot of biodiversity. We also explained what kind of biological studies are conducted historically there. We added the following lines in the text.

Lines 85-88:” Historically, the Roscoff Biological Station was established in this area because of the high species diversity. Indeed, coastal plankton [23,24], algae [25–27] and macro-organisms [28–30] have all now been intensively studied in the Roscoff region where they show high biodiversity and habitat variability.”

Reviewer 1: Line 93: These kinds of words are insufficiently precise to use in any study. Delete or be specific.

AUTHORS: This has been modified accordingly.

Lines 88-90:” This area is also characterised by human influences such as tourism and fishing activities, for which the impact on the meiobenthos remains poorly described.”

Reviewer 1: Line 177: What kind of grinder?

AUTHORS: The grinder is model mixer MM 400, this information has been added to the text

Reviewer 1: Line 226: ??

AUTHORS: Following the remark of Rev1, we added to the MS the following text: “Thus, images were processed on Fiji software, an open-source platform for biological-image analysis” (Line 223 reference 41: Schindelin J, Arganda-Carreras I, Frise E, Kaynig V, Longair M, Pietzsch T, et al. Fiji: an open-source platform for biological-image analysis. Nature methods. 2012;9: 676–682.)

Reviewer 1: Line 230: ??

AUTHORS: Concerning this remark, please consider that Otsu is the author of the method of automatic threshold selection for picture segmentation used in this study (Line 228 reference 42: Ostu N. A threshold selection method from gray-level histograms. IEEE Trans SMC. 1979;9: 62)

Reviewer 1: Line 248: Confusing as stated. Not easy to understand the differences.

AUTHORS: We thank Rev1 for this remark. This sentence needed a clarification. The text has been changed as follows: 

Lines 244-246: “Most environmental parameters showed little or no seasonal variation, except in August, when temperatures became warmer and oxygen more depleted: up to 11 °C and 132 µM, respectively (Table 1).”

Lines 322-323: “Temperature was also higher in the bay and harbour stations (Table 1).”

Reviewer 1: Line 253: Decreases?

AUTHORS: We thank Rev1 for this question, we clarified the text as follows:

Lines 248-249:” Salinity only showed slight variations apart from an abrupt decrease at station 5 in May and August.”

Lines 253-254:” In May, salinity decreased while oxygen, pH, and Chl a concentration increased drastically.”

Lines 319-321:” The salinity was homogeneous among almost all the sites except the estuarine station 5 and station 3 (down to 10–15) influenced by terrestrial freshwater (Table 1).”

Reviewer 1: Lines 269-270: I guess you mean May was second highest?

AUTHORS: Yes, the Rev1 is right. We corrected this mistake.

Lines 253-254:” In May, salinity decreased while oxygen, pH, and Chl a concentration increased drastically.”

Reviewer 1: Lines 323: CAN’T ALL THESE DESCRIPTIONS BE ELIMINATED BY SIMPLY REFERRING TO TABLE 1? IF THE DIFFERENCES MEAN ANYTHING LATER, THE TABLE MAY BE REFERRED TO

AUTHORS: We agree with this proposition. We referred to table 1 and modified the text according to the reviewers' suggestions to make it easier to understand.

Reviewer 1: Line 482: How many?

AUTHORS: Rev1 is right, we modified the text in the results to reply to this question. 

Lines 248-249:” Salinity only showed slight variations apart from an abrupt decrease at station 5 in May and August.”

Lines 319-321: “The salinity was homogeneous among almost all the sites except the estuarine station 5 and station 3 (down to 10–15) influenced by terrestrial freshwater (Table 1).”

Reviewer 1: Line 511: Unclear. Do you mean sediment was washed into or out of the sampling site? You surely do not mean washing the sediment, like with soap and water. You mean transporting sand and organisms, right?

AUTHORS: We thank Rev1 for these questions. Concerning his/her questions, the answer is yes, we meant transporting sand and organisms by storms or high hydrodynamics. This has been clarified in the text. 

Lines 365-367:” The most probable hypothesis would thus be that climatic events (i.e. storms with strong wind gusts) have periodically disturbed the ecosystem, resulting in the washing or mixing of sediments [46].”

Reviewer 1: Line 513: Competition for what? Space, food, less predation, or what?

AUTHORS: We thank Rev1 for these questions. It has been shown that amensalism can occur between copepods and foraminifera. This would be more a competition for food (line 373 reference 47: Chandler GT. Foraminifera may structure meiobenthic communities. Oecologia. 1989;81: 354–360. doi:10.1007/BF00377083). We clarified this part in the text as follows. 

Lines 371-373:” This decrease of faunal abundance could also be explained by amensalism or competition between foraminifera and other meiofaunal taxa (i.e. nematodes, copepods, [47]).”

Reviewer 1: Line 522: Do you mean flows and turbulence?

AUTHORS: Yes, in this area we have a stronger current due to the channel and this can be seen by the grain size with larger diameter grains.

Lines 449 -450: “This species is also known to be associated with areas with high hydrodynamics and has a preference for areas with marine channel influence [55,62].”

Reviewer 1: Lines 565-582: Most of this is speculation. You can rephrase it as alternative hypotheses with whatever evidence supports them. 

AUTHORS: The Rev1 is right, we were too far with hypothesis. The hypothesis on C. lobatulus has been deleted and the rest of the paragraph reworded.

Reviewer 1: Line 574: Drastic? How? Very high, sudden, or what? Delete or find another word.

AUTHORS: This has been modified and replaced by “markedly associated“.

Lines 473-476: “The species Reophax scorpius was markedly associated with the increase of temperature and would thus represent a good indicator to monitor large environmental disturbances induced by global warming at a local scale.”

Reviewer 1: Line 650: REFEREMCE ?

AUTHORS: We carefully checked reference list. This is reference 6. It is quoted on line 56

Reviewer 1: Line 718: REFERENCE ??

AUTHORS: We carefully checked reference list. This is reference 26. It is quoted on line 88 of the first version, line 86 in the amended version.

Reviewer 2:

Reviewer 1: Dachè et al made a sincere effort to understand the response of benthic foraminifera in the coastal settings of Roscoff Aber Bay. The manuscript is poorly drafted and needs a significant improvement. I suggest authors to overhaul the result and discussion section. First start with inter-season, then move to intra-season details. Although, authors have generated a large number of dataset from the region but fails to provide a comprehensive finding and not able to marks the relationship between benthic foraminifera abundance and diversity with ambient ecological parameters.

AUTHORS: We thank Rev2 for his/her comments. We agree that the ms needed an important rewriting and improvement. We followed carefully all the comments and suggestions made by Rev2 and we hope that the amended version is improved and can fulfil the requests made by Rev2.

Reviewer 2: Line 21: remove “good”.

AUTHORS: As requested by Rev2 this word has been removed

Reviewer 2: Line 79-81: ‘It is……….its salinity’. Provide references.

AUTHORS: Modified and quoted in the text: “line 79 reference 21: Hourdez S, Boidin-Wichlacz C, Jollivet D, Massol F, Rayol MC, Bruno R, et al. Investigation of Capitella spp. symbionts in the context of varying anthropic pressures: First occurrence of a transient advantageous epibiosis with the giant bacteria Thiomargarita sp. to survive seasonal increases of sulfides in sediments. Science of the Total Environment. 2021;798: 149149.”

Reviewer 2: Line 103: remove ‘temporal’

AUTHORS: As requested by Rev2 this word has been removed

Lines 97-99: “In this context, we analysed the spatial and seasonal distribution of living benthic foraminifera from contrasting habitats impacted by natural and anthropogenic environmental changes, with the aim of identifying bio-indicator species.”

Reviewer 2: Line 123: Mark the freshwater stream in figure 1 clearly.

AUTHORS: We thank Rev2 for this remark. The figure has been edited accordingly

Reviewer 2: Line 137-138: It is hard to distinguished that authors used only living benthic foraminifera or total benthic foraminifera population including dead shells (in the top 5 cm). Needs to mention clearly.

AUTHORS: We thank Rev2 for this comment. The text was not sufficiently clear. In the amended version of the MS, we clarified that all biological data are based on living specimens only. 

Lines 130-132: ”In this study, only a single core was used to determine living benthic foraminifera (by phloxine B staining) [33].”

Reviewer 2: Line 157-160: Please provide accuracy of conductivity, pH and oxygen measurements.

AUTHORS: Modified as requested

Ligne 151-156: “At all stations, salinity parameters were measured with an LF 340 handheld conductivity meter with a standard TetraCon 325 conductivity cell (Measuring Range 1 µS/cm - 2 S/cm) and pH parameters were measured with a WTW pH 3310 sensor (accuracy ± 0.005). Temperature and oxygen measurements were made with a Oxygen Optode 3830 (temperature accuracy ±0.05°C and O2-concentration accuracy < 8 μM or 5%, whichever is greater).”

Reviewer 2: Line 187: Sulfanilic acid was used as the elemental standard. Why acid was used as a standard for sediment samples, where standard and samples values deviate with a large difference.

AUTHORS: We thank Rev2 for this comment. The explanation is that we did not find commercially available and certified sediment standards for elemental analysis. This shortcoming is compensated for by the use of an "in-house" Mediterranean sediment standard, which is not certified but ensures that the values measured are repeatable.

Reviewer 2: Line 206-207: ‘4% formalin.....foraminiera’. cite reference.

AUTHORS: Reference 33 Line 203: “Mason WT, Yevich PP. The Use of Phloxine B and Rose Bengal Stains to Facilitate Sorting Benthic Samples. Transactions of the American Microscopical Society. 1967;86: 221–223. doi:10.2307/3224697”

Reviewer 2: Line 232: put space between % and area.

AUTHORS: Modified as requested.

Reviewer 2: Line 252: ‘pH….sites. Remove the sentence. 7.2 to 8.8 is not a little variation by any mean.

AUTHORS: We thank Rev2 for this suggestion. The comment has been taken into account and modified as requested.

Lines 246-248: “The pH showed local variations, such as strong increases in May and August at stations 2, 3 4, 7 and 8 and in August only at station 6.”

Lines 321-322: “The pH values were higher in the bay and harbour stations (~7.8–8.2) than in the channel stations (~7.6).”

Reviewer 2: Results: Results should be written first inter-season wise, followed by intra-seasonal variations. It is very hard to follow it in its current form.

AUTHORS: We thank Rev2 for this proposition. We changed the structure of the results and discussion, following this suggestion. 

Reviewer 2: Line 269: ‘Oxygen concentration was highest in November’? at which station? All of them? If yes, then mention. Also provide values for it rather than writing highest and lowest. Correct it throughout the result section.

AUTHORS: All results have been modified using a new structure and values are provided in the text. 

Lines 244-246:” Most environmental parameters showed little or no seasonal variation, except in August, when temperatures became warmer and oxygen more depleted: up to 11 °C and 132 µM, respectively (Table 1).”

Lines 324-327:” The opposite trend was observed regarding oxygen availability, with minima observed at the harbour station 6 (21 µM) and estuarian station 5 (17 µM) and maxima at bay stations 3 and 4 (188–264 µM).”

Reviewer 2: Line 287: δ13C, δ15N, δ34S. Make mass value superscript.

AUTHORS: Modified as requested.

Reviewer 2: Line 309-311: The δ34S………seasons. It does not belong to result section. Remo

---

## [Decision Letter · Decision Letter 1]

13 Aug 2024

Benthic Foraminifera as bio-indicators of natural and anthropogenic conditions in Roscoff Aber Bay (Brittany, France)

PONE-D-24-06208R1

Dear Dr. Dache,

We’re pleased to inform you that your manuscript has been judged scientifically suitable for publication and will be formally accepted for publication once it meets all outstanding technical requirements.

Kind regards,

Marcos Rubal García, PhD

Academic Editor

PLOS ONE

Additional Editor Comments (optional):

Reviewers' comments:

Reviewer's Responses to Questions

**Comments to the Author**

1. If the authors have adequately addressed your comments raised in a previous round of review and you feel that this manuscript is now acceptable for publication, you may indicate that here to bypass the “Comments to the Author” section, enter your conflict of interest statement in the “Confidential to Editor” section, and submit your "Accept" recommendation.

Reviewer #2: All comments have been addressed

2. Is the manuscript technically sound, and do the data support the conclusions?

Reviewer #2: Yes

3. Has the statistical analysis been performed appropriately and rigorously? 

Reviewer #2: Yes

4. Have the authors made all data underlying the findings in their manuscript fully available?

Reviewer #2: Yes

5. Is the manuscript presented in an intelligible fashion and written in standard English?

Reviewer #2: Yes

6. Review Comments to the Author

Reviewer #2: (No Response)

7. PLOS authors have the option to publish the peer review history of their article (what does this mean?). If published, this will include your full peer review and any attached files.

Reviewer #2: **Yes: **Dharmendra Pratap Singh

---

## [Editor Report · Acceptance letter]

10 Oct 2024

PONE-D-24-06208R1 

PLOS ONE

Dear Dr. Daché, 

I'm pleased to inform you that your manuscript has been deemed suitable for publication in PLOS ONE. Congratulations! Your manuscript is now being handed over to our production team.

Kind regards, 

on behalf of

Dr. Marcos Rubal García 

Academic Editor

PLOS ONE